# Utilizing wearable sensors for continuous and highly-sensitive monitoring of reactions to the BNT162b2 mRNA COVID-19 vaccine

Yftach Gepner[1,7], Merav Mofaz[2], Shay Oved[2], Matan Yechezkel[2], Keren Constantini[1], Nir Goldstein[1], Arik Eisenkraft[3,4], Erez Shmueli [2,5,7] & Dan Yamin [2,6,7 ✉]

## Abstract

**Background** Clinical trial guidelines for assessing the safety of vaccines, are primarily based on self-reported questionnaires. Despite the tremendous technological advances in recent years, objective, continuous assessment of physiological measures post-vaccination is rarely performed.

**Methods** We conducted a prospective observational study during the mass vaccination campaign in Israel. 160 participants >18 years who were not previously found to be COVID-19 positive and who received the BNT162b2 COVID-19 (Pfizer BioNTech) vaccine were equipped with an FDA-approved chest-patch sensor and a dedicated mobile application. The chest-patch sensor continuously monitored 13 different cardiovascular, and hemodynamic vitals: heart rate, blood oxygen saturation, respiratory rate, systolic and diastolic blood pressure, pulse pressure, mean arterial pressure, heart rate variability, stroke volume, cardiac output, cardiac index, systemic vascular resistance and skin temperature. The mobile application collected daily self-reported questionnaires on local and systemic reactions.

**Results** We identify continuous and significant changes following vaccine administration in nearly all vitals. Markedly, these changes are observed even in presumably asymptomatic participants who did not report any local or systemic reaction. Changes in vitals are more apparent at night, in younger participants, and in participants following the second vaccine dose.

**Conclusion** the considerably higher sensitivity of wearable sensors can revolutionize clinical trials by enabling earlier identification of abnormal reactions with fewer subjects.

## Plain language summary

The safety of vaccines in clinical trials is primarily determined by participants completing self-reported questionnaires. We monitored various indicators of participant's health using a chest-patch sensor in 160 participants before and after receiving the BNT162b2 COVID-19 (Pfizer BioNTech) vaccine. Participants were also asked to self-report their health via a mobile phone app. We observed significant changes in health indicators following vaccine administration. Changes were seen by chest patch sensor in both participants who did and did not report changes via the mobile phone app. Three days following vaccination, participant health indicators returned to the levels observed the day before vaccination in both groups. Using wearable sensors could potentially improve clinical trials by enabling earlier identification of abnormal reactions.

[1] Department of Epidemiology and Preventive Medicine, School of Public Health, Sackler Faculty of Medicine, and Sylvan Adams Sports Institute, Tel-Aviv University, Tel-Aviv, Israel. [2] Department of Industrial Engineering, Tel-Aviv University, Tel-Aviv, Israel. [3] The Institute for Research in Military Medicine, the Hebrew University Faculty of Medicine, Jerusalem, Israel. [4] Chief Medical Officer, Biobeat Technologies Ltd., Petah-Tikva, Israel. [5] MIT Media Lab, Cambridge, MA, USA. [6] Center for Combatting Pandemics, Tel-Aviv University, Tel-Aviv, Israel. [7] These authors contributed equally: Yftach Gepner, Erez Shmueli, Dan Yamin. ✉email: dan.yamin@gmail.com

Vaccination is widely accepted as the most prominent measure in the fight against COVID-19, posing the greatest hope for ending this major global health pandemic and related economic crisis[1,2]. Consequently, an unprecedented international effort by private and public institutions alike was directed at accelerating the traditionally lengthy vaccine-development process[3–5]. On 2 December 2020, less than a year from the pandemic outbreak, the first vaccine, BNT162b2 mRNA (Pfizer-BioNTech), was granted an Emergency Use Authorization (EUA) by the UK Medicines and Healthcare products Regulatory Agency (MHRA)[6]. This initial authorization was followed by rapid authorizations for emergency use in several countries, with the US Food and Drug Administration (FDA) among the first to do so[7]. The promising BNT162b2 vaccine was demonstrated to have 95% efficacy in preventing symptomatic COVID-19 in clinical trials[8], and 92% efficacy in a nationwide mass vaccination[9].

Safety data from a randomized, controlled trial suggests a favorable safety profile for the BNT162b2 vaccine[8]. Specifically, the local and systemic reactions reported during the first seven days after vaccination were mostly mild to moderate, with a median onset of 0–2 days after vaccine administration and a median duration of 1–2 days. The most frequently reported reactions were fatigue, headache, muscle pain, chills, joint pain, and fever. The incidence of serious adverse events was low and was similar between vaccine- and placebo-treated participants. The safety of the new vaccine over a median of two months post-vaccination was similar to that of other viral vaccines. A considerable fraction of the participants did not report any reaction or adverse event. Likewise, several other vaccine candidates, including ChAdOx1 nCoV-19 (Oxford/AstraZeneca) and mRNA-1273 (Moderna), received EUAs following similar encouraging safety results in randomized, controlled trials[10–12].

Nevertheless, concerns regarding potential adverse effects from vaccines have recently led to the suspension of the ChAdOx1 nCoV-19 vaccination campaigns in several European countries[13] and may have reinforced the public hesitancy towards COVID-19 vaccines. These concerns underscore the importance of extracting as much information as possible from clinical trials. However, to date, clinical trial guidelines for assessing the safety of vaccines, including the FDA criteria[14], are primarily based on subjective, self-reported questionnaires. Despite the tremendous technological advances in recent years, objective, continuous assessment of physiological measures post-vaccination is rarely performed.

Here, we evaluated the short-term effects of the BNT162b2 COVID-19 vaccine on physiological measures. We followed a cohort of 160 participants who received the second dose of the BNT162b2 vaccine for 96 hours, from 24 h prior vaccine administration until 72 h after the inoculation. Participants were fitted with a chest-patch sensor that monitored objectively and continuously 13 different physiological indicators. Additionally, a dedicated mobile application was used to record daily self-reported questionnaires on local and systemic reactions. We identified considerable changes in chest-patch indicators during the first 48 h post-vaccination also in this group of presumably asymptomatic participants. These measures returned to the levels observed during the day prior vaccination in both groups, further supporting the safety of the vaccine. Our findings underscore the importance of accounting for objective technological advances in clinical trials to more accurately understand the invisible impact of the vaccine on our respiratory, cardiovascular, and hemodynamic systems. Extracting as much information as possible from clinical trials, particularly during an emergency, is crucial for a more comprehensive determination of vaccine safety.

## Methods

**Study design and participants.** Our study includes a prospective cohort of 160 participants who were not previously found to be COVID-19 positive and received the second dose of the BNT162b2 mRNA COVID-19 vaccine between 1 January 2021 and 13 March 2021. This sample size of >150 participants was chosen to ensure a sufficient amount of participants will present substantial reactions (such as fever)[8]. Specifically, as reported reactions were considerably more severe after the second dose compared to the first[8], we focused on the second dose of vaccine. Out of the 160 participants in this study, 90 (56.25%) were women and 70 (43.75%) were men. Their age ranged between 21 and 78 years, with a median age of 40. Participants were equipped with a chest-patch sensor and were monitored for a period of four days, starting one day prior to vaccine administration. In addition, participants installed a dedicated mobile application and were requested to fill in a daily questionnaire, starting one day prior to the inoculation, for a period of 15 days. For each participant, the measurements levels were compared to the levels observed on the 24-h period prior to vaccination.

In order to recruit participants and ensure they complete all the study's requirements, we hired a professional survey company. Potential participants were recruited through advertisements in social media, online banners, and word-of-mouth. The survey company was responsible for guaranteeing the participants met the study's requirements, in particular, that they agreed to wear the chest-patch sensor and fill in the daily questionnaires.

Participants were met in person, roughly 24 h prior to vaccination, and received a detailed explanation about the study, after which they were requested to sign an informed consent form. Then, participants were asked to complete a one-time enrollment questionnaire and install two applications on their mobile phones: an application that passively collects data from the chest-patch sensor and the PerMed application, allowing participants to fill the daily questionnaires.

To better understand the effects of the second vaccine dose on physiological measures, we also monitored 24 participants, with an identical procedure, when receiving their first vaccination dose.

**The chest-patch sensor.** The photoplethysmography (PPG)-based chest monitors purchased and used in this study collects the following 13 indicators of vital signs: heart rate (bpm), blood oxygen saturation (%), respiratory rate (br/min), systolic and diastolic blood pressure (mmHg), pulse pressure (mmHg), mean arterial pressure (mmHg), heart rate variability (HRV—the calculation is based on a time-domain method in which the root mean square of successive RR interval differences are measured for a segment of ten beats and are presented in %), stroke volume (mL/beat), cardiac output (L/min), cardiac index (L/min/m2), systemic vascular resistance (SVR) (dynes·sec·cm$^{-5}$), and skin temperature (c). Together, these indicators of vital signs provide an accurate and comprehensive assessment of the respiratory, cardiovascular, and hemodynamic. The chest patch sensor received FDA clearance for measures of heart rate, blood oxygen saturation, and systolic and diastolic blood pressure (clearance number K190792) and CE Mark approval for all 13 measures (CE 2797) (Biobeat Technologies Ltd), (see also a technological validation study for blood pressure[15]). The sensor tracks vital signs derived from changes in the pulse contour, following calibration using an approved non-invasive, cuff-based device, and is based on Pulse Wave Transit Time (PWTT) technology, combined with pulse wave analysis (PWA) (see, for example, refs. [16,17]). To increase its validity, the device has an inherent component that prevents showing values if the movement has crossed a pre-set threshold. To the best of our knowledge, this is the only cleared

wearable wireless medical-grade device to provide all these measurements. As the chest patch sensors' battery typically lasts for 4–5 days, participants were asked to remove the chest patch sensors 3 days after vaccination. Accordingly, the sensor continuously collected data at 10-min intervals for the entire duration of the 96-h experiment.

**PerMed mobile application**. Developed originally to support the PerMed study[18], the PerMed mobile application allows participants to fill the daily questionnaires. The daily questionnaire we used included questions about clinical symptoms from a closed list of local and systemic reactions observed in the BNT162b2 mRNA Covid-19 clinical trial[8], with an option to add other symptoms as free text.

In order to improve the quality and reliability of the data and to ensure its continuous collection, we applied the following two measures: (1) Participants who did not complete the daily questionnaire by 7 p.m. received a notification in their mobile app to fill the questionnaire; (2) We developed a dedicated dashboard that helped us identify when participants did not fill in the daily questionnaires. Those participants were contacted by the survey company and were encouraged to cooperate better.

**Statistics and reproducibility**. Before analyzing the data, we performed several preprocessing steps. With regard to the daily questionnaires, in cases where participants filled in the daily questionnaire more than once on a given day, only the last entry for that day was considered, as it was reasoned that the last one likely best represented the entire day. Self-reported symptoms that were entered as free text were manually categorized. With regard to the chest-patch indicators, data were first aggregated per hour (by taking the mean value). Then, to impute missing values, we performed a linear interpolation. To validate the consistency of our findings, we split the data into two groups in a balanced fashion in terms of their age group and gender. One group contained 80%, and the other one contained the other 20%. The final analyses presented are based on the entire sample. All data and code required to reproduce the results reported in this paper are available in the GitHub repository[19].

To examine the changes in each chest-patch indicator over the 72 h post-vaccination compared with the 24-h period prior to vaccination, we performed the following steps. For each indicator, for each hour h of the 72 h post-vaccination, we calculated for each individual the mean value in the 5-h sliding window: [h−4, h−3, h−2, h−1, h]. Then, we calculated the relative change in the percentage of this value compared to the corresponding 5-h window in the day prior to vaccination. Finally, we calculated the mean value for hour h over all 160 participants, as well as the 90% confidence interval, corresponding to a significance level of 0.05 in a one-sided t-test (Fig. 1). We performed a similar analysis for the 24 participants who received the first vaccine dose (Supplementary Fig. S1).

Next, for each of the 3 days after the vaccination, we calculated the percentage of participants who reported new local or systemic reactions compared to their reports on the day prior vaccination (Fig. 2). For each reaction, a 90% confidence interval was calculated assuming a beta distribution, with parameter $\alpha$ corresponding to the number of participants reporting that reaction plus one (i.e., "successes"), and parameter $\beta$ corresponding to the number of participants who did not report that reaction plus one (i.e., "failures").

Finally, we examined the difference between symptomatic and asymptomatic participants with regard to changes in the chest-patch indicators, stratified by the number of days post-vaccination (1–3) and part of the day (day or night) (Fig. 3).

For a given day post-vaccination, symptomatic participants were defined as those who reported at least one reaction on that day that they did not report the day prior vaccination. Asymptomatic individuals were defined as those who reported no reactions on that day. We defined nighttime as the time interval between 12 a.m. and 7 a.m. and daytime between 7 a.m. and 12 a.m. This day-night definition is consistent with the observed movement patterns of the participants throughout the study. For each participant, we calculated the mean indicator value for each day and part of the day post-vaccination. Then, we calculated the relative change in percentages of these values compared to their corresponding values in the day prior vaccination. Next, we calculated the mean values of symptomatic participants and asymptomatic participants, as well as their corresponding 90% confidence intervals using a t distribution. Finally, unequal variances t-tests were used to evaluate the differences between symptomatic and asymptomatic participants.

**Ethical approval**. Before participating in the study, all subjects were advised, both orally and in writing, as to the nature of the study and gave written informed consent to the study protocol, which was approved by the Tel Aviv University Institutional Review Board (0002522-1). All data were de-identified and no personally identifiable information was gathered.

**Reporting summary**. Further information on research design is available in the Nature Research Reporting Summary linked to this article.

## Results

Between 1 January 2021 and 13 March 2021, a total of 166 participants were recruited. Among them, 160 participants completed the trial, and their data were analyzed; five participants left the trial early, and one participant was provided with a malfunctioning chest-patch sensor. Among these participants, 56.25% were females, 13.75% were obese (body-mass index of at least 30.0), 18.12% had high blood pressure, and 20.00% had at least one comorbidity. The median age was 40 years and 10.63% of participants were older than 59 years of age (Table 1).

Within the first 48 h post-vaccination, we identified significant changes in nearly all 13 chest-patch indicators compared to the levels observed in the day prior vaccination (Fig. 1, Supplementary Table S1, and Fig. S2). For example, at their peak, the heart rate increased by 9.85% (90% CI, 7.71 to 11.99%), the systolic blood pressure increased by 3.91% (90% CI, 2.97 to 4.87%), and the diastolic blood pressure increased by 3.78% (90% CI, 2.82 to 4.74%). By contrast, we observed no significant differences in blood oxygen saturation (Fig. 1h). Following the initial 48 h, these changes faded, with measurements returning to their levels observed on the day prior vaccination.

Focusing on self-reported reactions, we find consistent trends with those observed in the BNT162b2 mRNA vaccine clinical trial (Fig. 2 and Supplementary Fig. S3). Specifically, the most frequent reactions reported via the self-reported questionnaires collected from the mobile application were fatigue, headache, muscle pain, fever, and chills. Consistent with our analysis of the chest-patch indicators, participants reported most of the reactions during the first 2 days post-vaccination, followed by a sharp decline in reporting on the third day and nearly a complete halt within 14 days post-vaccination. Importantly, almost half of the participants (48.5%) did not note any local or systemic reaction.

Importantly, we identified significant changes in chest-patch indicators during the first 2 days post-vaccination also in presumably asymptomatic participants (Fig. 3 and Supplementary Fig. S4). Specifically, during the daytime of the first 2 days post-

**Table 1 Characteristics of the participants.**

| Characteristic | Number of participants (%) ($N = 160$) |
|---|---|
| **Gender** | |
| Male | 70 (43.75%) |
| Female | 90 (56.25%) |
| **Age group** | |
| 18–39 years | 91 (56.87%) |
| 40–59 years | 52 (32.50%) |
| ≥60 years | 17 (10.63%) |
| **Body-mass index* (kg/m$^2$)** | |
| <30.0 | 128 (80.00%) |
| ≥30.0 | 22 (13.75%) |
| Unspecified | 10 (6.25%) |
| **High blood pressure measured at the day before vaccination (Systolic >140 mmHg or Diastolic >90 mmHg)**** | |
| No | 131 (8.88%) |
| Yes | 29 (18.12%) |
| **Household size** | |
| ≤2 | 67 (41.87%) |
| >2 | 83 (51.88%) |
| Unspecified | 10 (6.25%) |
| **Self-reported illness in the past 30 days** | |
| No | 104 (65.00%) |
| Yes | 46 (28.75%) |
| Unspecified | 10 (6.25%) |
| **Comorbidities** | |
| No | 128 (80.00%) |
| Hypertension | 11 (6.87%) |
| Diabetes | 7 (4.37%) |
| Heart disease | 3 (1.87%) |
| Chronic lung disease | 5 (3.12%) |
| Renal failure | 1 (0.62%) |
| Unspecified | 12 (7.50%) |

* The body-mass index is the weight in kilograms divided by the square of the height in meters.
** Blood pressure was measured a day before inoculation.

vaccination, nearly all 13 chest-patch indicators significantly changed not only for symptomatic participants (i.e., those who reported at least one local or systemic reaction) but also for asymptomatic participants (i.e., those who did not report any local or systemic reaction). During the night hours, among the asymptomatic participants, significant changes were observed mainly on the first-night post-vaccination. Conversely, among the symptomatic participants, significant changes were observed on both the first- and second-nights post-vaccination. Moreover, the changes observed for symptomatic participants were found to be significantly higher than those of asymptomatic individuals in 9 out of the 13 chest-patch indicators during the first-day post-vaccination and in 7 out of the 13 chest-patch indicators during the first-night post-vaccination (unequal variances t-test, p value <0.05).

We also analyzed the changes in chest-patch indicators, stratified by age group, gender, and vaccine dose. Participants 60 years old and above exhibited milder, albeit not significant, changes than those below 60 years old in nearly all chest-patch indicators (Supplementary Figs. S5, S6, S1). We found no differences between men and women in chest-patch indicators. In contrast to the second vaccine dose, examining the subset of 24 participants who were monitored when receiving their first vaccine dose revealed no difference in vitals during the first 48 h post-inoculation (Supplementary Fig. S1).

## Discussion

Our key findings suggest that multiple physiological measures significantly change following BNT162b2 COVID-19 vaccine administration in both participants who reported and those who did not report local and systemic reactions. Within three days from vaccination, these measures returned to the levels observed the day before vaccination in both groups, further supporting the safety of the vaccine.

We identified several aspects that strengthen the short-term safety of the BNT162b2 vaccine. First, we found that all the physiological measures returned to their levels observed on the day prior vaccination within three days from vaccination. Second, we observed no change in oxygen saturation levels, indicating that major adverse health consequences are less likely. Third, reports of local and systemic reactions declined considerably on the third day following vaccination. Thus, our study quells, in part, concerns raised by those hesitant to be vaccinated on the grounds of potential adverse vaccination consequences by demonstrating in an objective measurement that such consequences are likely to fade after a few days.

We identified a considerable decline in systemic vascular resistance (SVR) during the first 2 days after inoculation. Such a decline is typically regarded as an indication of inflammatory response. Hematology studies performed by Pfizer and reported in the VRBPAC document[20], suggest that the most commonly observed changes were transient grade 1 or 2 decreases in lymphocytes, 1 to 3 days after the first vaccine dose. These decreases returned to levels observed on the day prior vaccination within 6–8 days of the first dose. The authors note that RNA vaccines are known to induce type I interferon, which regulates lymphocyte recirculation and are associated with transient migration and redistribution of lymphocytes. This rapid rebound of lymphocytes supports the notion that they were not depleted, but temporarily migrated out of the peripheral blood and subsequently reentered the bloodstream by the next assessment time. Although these hematology observations were after the first dose, the change in vascular resistance could provide compelling support for the idea of migration of lymphocytes out of circulation. However, our study did not include hematology tests. Though SVR is regarded as an indication of the inflammatory response, the fact it is counterbalanced with cardiac output, and the lack of other inflammatory indicators, including skin temperature, show that the decrease is not necessarily directly related in this situation to inflammation. This could be further looked at in future studies, either with this specific vaccine or with any other, emphasizing the need to exhaust all available data in clinical research to detect—as early as possible—any side effects among all volunteers.

Clinical trial guidelines for assessing the safety of vaccines, including the FDA criteria, are primarily based on subjective, self-reported questionnaires. Using a real-time remote patient monitoring platform that can provide multiple objective physiological parameters has significant implications for clinical studies of all phases in learning about adverse events, safety, and choosing the ideal treatment protocol. Moreover, it has been recently suggested that reactions caused by the COVID-19 vaccine are a byproduct of a short burst of IFN-I generation concomitant with the induction of an effective immune response[21]. Thus, future studies should evaluate the association between physiological reactions and vaccine effectiveness.

Several recent studies highlighted the pivotal role of heart rate variability for the detection of COVID-19 infection[22–26]. Specifically, one study showed that a combination of measures from smartwatches, including HRV, can be utilized to detect COVID-19 during the presymptomatic phase[23]. Although significantly changed, we did not observe a robust signal from the HRV measure following vaccination. As HRV is reported to be an index of the influence of both the parasympathetic nervous

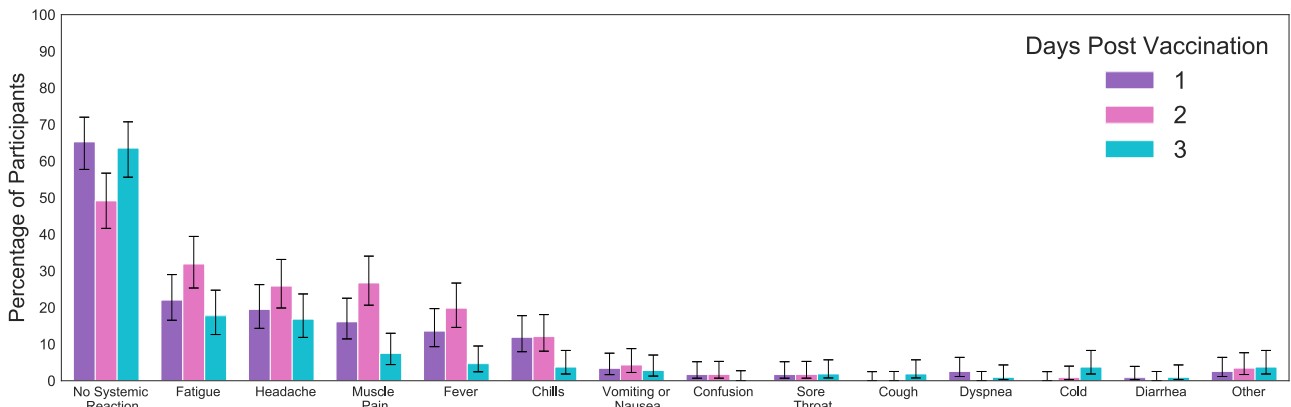

**Fig. 1 Percentage of change in chest-patch indicators compared to their levels observed on the day prior to vaccination.** Percentage of change in respiratory, cardiovascular, and physiological indicators recorded by the chest-patch sensor compared to their levels observed on the day prior vaccination: **a** skin temperature, **b** heart rate, **c** cardiac output, **d** systemic vascular resistance, **e** systolic blood pressure, **f** diastolic blood pressure, **g** respiratory rate, and **h** oxygen saturation. Mean values are depicted as solid lines, 90% confidence intervals are presented as shaded regions, and horizontal dashed lines represent no change compared to the levels observed on the day prior to vaccination. The analysis is based on $n = 160$ participants.

**Fig. 2 Local and systemic reactions reported by participants through the mobile application.** Error bars represent 90% confidence intervals. The number of participants who filled in the daily questionnaire on the first-, second-, and third-day post-inoculation was $n = 118$, 116, and 108, respectively.

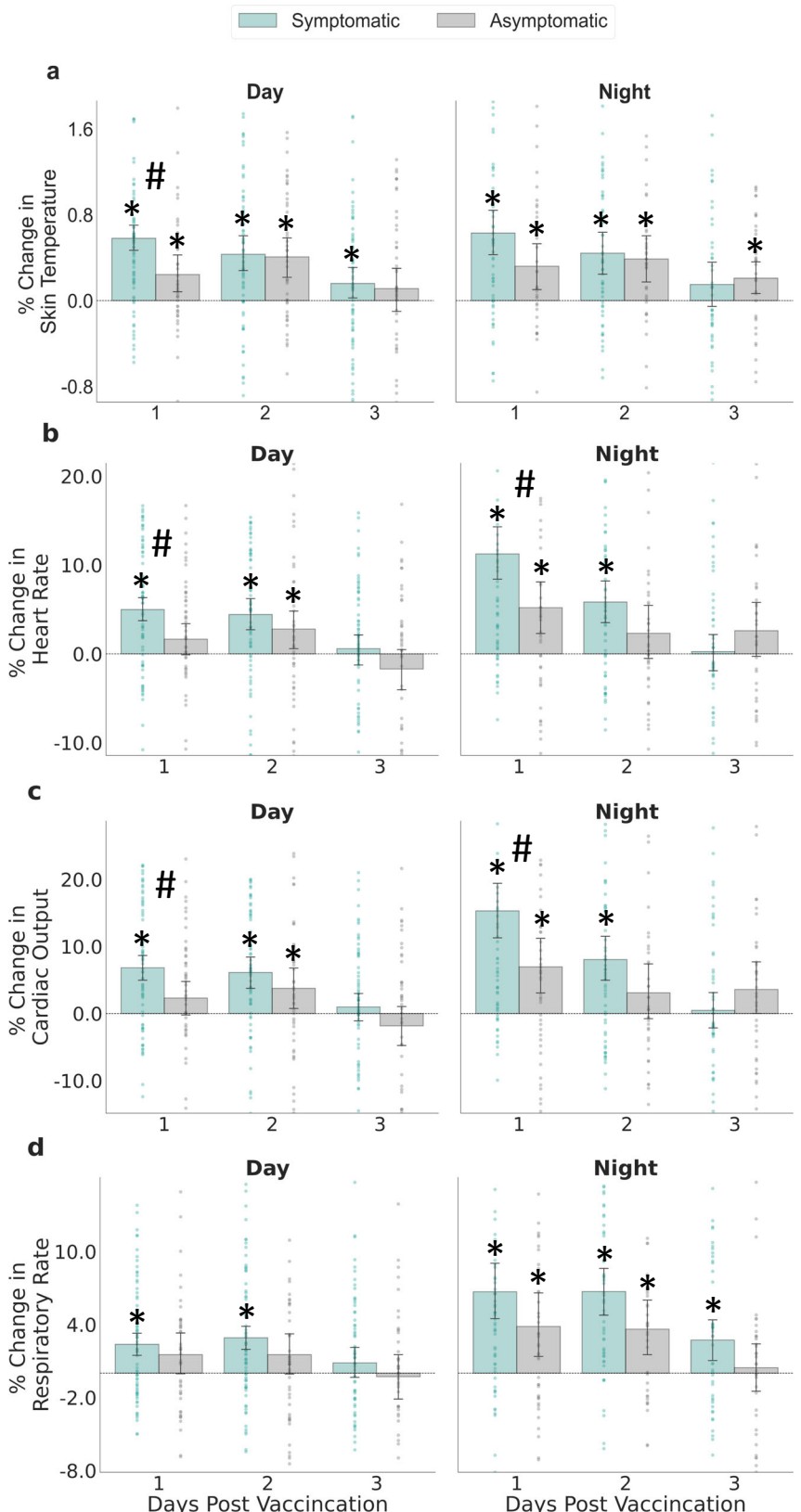

**Fig. 3 Percentage of change in chest-patch indicators for participants who reported at least one local or systemic reaction and those who reported no reaction during the daytime and the nighttime.** Percentage of change in chest-patch indicators for participants who reported at least one local or systemic reaction (symptomatic, $n = 80$ participants) and those who reported no reaction (asymptomatic, $n = 55$ participants) during the daytime and the nighttime: **a** skin temperature, **b** heart rate, **c** cardiac output, and **d** respiratory rate. Error bars represent 90% confidence intervals. Horizontal dashed lines represent no change compared to the levels observed on the day prior to vaccination. Significant differences compared to the measurements levels observed on the day prior to vaccination at a 0.05 level are marked with *. Significant differences between symptomatic and asymptomatic participants at a 0.05 level are marked with #.

system and the sympathetic nervous systems[27], it could be that vaccination affects these systems differently than infection.

Our study includes several limitations. First, our cohort includes only 160 participants and may not adequately represent the vaccinated population in Israel and elsewhere. However, compared to other studies monitoring vaccines with wearable sensors, and particularly COVID-19 vaccines, our sample size is relatively large. Nevertheless, despite this considerably small sample size, trends observed by the chest-patch sensor were found to be significant. Moreover, the proportion of those who reported local and systemic reactions and the type of reactions noted were similar to those observed in clinical trials[8]. Second, all participants received the BNT162b2 vaccine, which was the only vaccine used in Israel. Given the similarities in the extents and the types of local and systemic reactions observed between different COVID-19 vaccines[8,11,12], we believe our findings are likely to be qualitatively similar in other vaccine types. Third, the rich data gathered by the chest-patch sensor was recorded for a relatively short period, 4 days. However, our findings from both the chest-patch sensor and the daily questionnaire (which was collected for a more extended period, i.e., 14 days) reveal that the vast majority of local and systemic reactions faded within 2 days. Fourth, we did not explicitly control for the effects of the clinical trial setting (i.e., participating in a trial, wearing a chest-patch sensor, potential concerns from the vaccine, etc.). While one may argue that the observed changes may be an artifact of the trial setting, we would expect to find these changes during the first vaccine dose as well. However, since we found no differences in most vitals in the subset of participants who received their first dose, we believe the changes observed after the second dose arises from an actual reaction to the vaccine.

About half of the participants did not report any local or systemic reaction following vaccination, which is consistent with previous reports from the clinical trial[8]. Yet, we show that both symptomatic and asymptomatic participants had substantial objective physiological changes regardless of their subjective reports. Hence, our work underscores the importance of obtaining objective physiological data in addition to self-reported questionnaires when performing clinical trials, particularly in those conducted in very short time frames.

Whereas self-reported trends are widely described in the scientific literature, no study or vaccine clinical trial has reported the comprehensive effects of the COVID-19 vaccine on physiological measures. In fact, current US FDA, and European Medicines Agency (EMA) guidelines for assessing the safety of newly developed vaccines are primarily based on subjective, self-reported questionnaires. Our findings should encourage public health officials and regulatory agencies to include the analysis of objective measures, in addition to self-reported questionnaires, as part of the evaluation of clinical trials. Though not available in the past, current digital wearable health technologies could offer simple platforms for obtaining physiological measures, adding invaluable objective, non-biased information. Thus, as part of the future process of vaccine approval, it is essential to combine objective and remote medical-grade monitors in clinical studies.

## Data availability
Source datasets of the results reported in this paper are available in the GitHub repository, https://github.com/permedtau/covid-vaccination-biobeat-paper.

## Code availability
Statistical code is available in the GitHub repository[19], https://github.com/permedtau/covid-vaccination-biobeat-paper. Statistical code was programmed using Python version 3.7, Numpy package version 1.19.5, Pandas package version 1.3.5, SciPy package version 1.7.1, Seaborn package version 0.9.0, and Matplotlib package version.

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

## Author contributions

Conception and design: D.Y., E.S., and Y.G., Collection and assembly of data: M.M. and S.O. Analysis and interpretation of the data: M.M., S.O., A.E., Y.G., E.S., and D.Y. Statistical expertize: D.Y., E.S., M.M., S.O., and M.Y. Drafting of the article: D.Y., E.S., Y.G., M.M., S.O., M.Y., K.C., N.G., A.E. Critical revision of the article for important intellectual content: E.S., D.Y., and Y.G. Final approval of the article: All authors. Obtaining funding: D.Y. and E.S.

## Funding

This work was supported by the European Research Council (ERC) project #949850 and the Israel Science Foundation (ISF), grant No. 3409/19, within the Israel Precision Medicine Partnership program. Keren Constantini was supported by a post-doctoral fellowship from the Tel Aviv University Center for Combatting Pandemics.

## Competing interests

Arik Eisenkraft is the Chief Medical Officer of Biobeat Technologies Ltd., the supplier of the chest-patch sensor. The remaining authors declare no competing interests.
