## [Peer Review File · Communications Medicine]

Reviewers' comments:

Reviewer #1 (Remarks to the Author):

Utilizing wearable sensors for objective, continuous and highly- sensitive monitoring of reactions to vaccines

Gepner et al. present a report of a wearable monitoring system that records physiologic data and the changes in those physiologic variables in response to the COVID-19 vaccine. The study design is sound and the manuscript does not have any major pitfalls. It is, however, disorganized as it reports results/discussion prior to the study population and study design. These should be re-organized.

While it is clear that the authors allocated a great deal of resources to this report, it is unclear to me how this would inform additional studies, from a physiologic perspective, outside of showing the feasibility of this study design/use of the wearable monitor. The physiologic metrics, which are the main strength of the paper and heavily analyzed in the results, need to be discussed in greater detail in regards to their importance and how this can be leveraged to either determine vaccine response, antibody titers or protective immunity.

The wellness and other subjective outcomes are fair to report, but they do not add much to the overall impact of the manuscript, aside from providing external validation of the much larger studies already published. The authors went to great lengths to collect this data with the mobile app, but even a profound impact on subjective outcomes may not even be clinically significant in the setting of an efficacious vaccine to end a global pandemic.

The next issue with this manuscript is the definition of the term "baseline." The device was worn for a total of 4 days (1 day prior to vaccination and 3 days after). The authors claim a "return to baseline," which is an inaccurate statement as they did not determine a true baseline for these physiologic metrics. This essentially nullifies any claims made about a return to baseline, as many of these factors fluctuate on a day-to-day basis and a "change from baseline" should be calculated based on at minimum a 30-day dataset (if not more).

A strength of the manuscript is its prospective nature and size (n=160). The latter is worth highlighting as it is likely one of the larger studies with vaccines and wearables, specifically the COVID vaccine. In addition, the descriptive analysis of the systemic and local reactogenicity was well done, and overall the figures are easy to follow.

Specific comments:

1. Recommend including the brand name of the vaccine that was used in this study for the lay person in the manuscript title. And please comment on whether the authors feel these results would be generalizable across all mRNA COVID-19 vaccines.
2. Title should specify that this report focuses on COVID-19 vaccination. While this may be applicable to all vaccines, it is not within the scope of this manuscript to make this claim (can be touched upon in the discussion).
3. Per above, please organize the manuscript to have the study population/design/methods etc before the results and discussion (should follow the outline of the abstract).
4. Was sleep duration a measured variable with the monitoring device? If so, please report.

5. The Day 0 and daily changes (days 1-3) in variables should be reported in table format to show the inter and intraindividual variability seen in the cohort.

6. Heart rate variability (HRV) has been studied as marker of autonomic function and perturbation, and correlations between HRV and vaccine responses have been reported in the literature. More recently, HRV has been studied in several reports related to COVID-19 vaccine (citations below). These should certainly be cited in this manuscript and commented upon in the discussion (specifically, why the authors think their continuous HRV measurements may not have shown as robust of a change than other reports in the literature).

1. Miller DJ, Capodilupo JV, Lastella M, Sargent C, Roach GD, Lee VH, et al. Analyzing changes in respiratory rate to predict the risk of COVID-19 infection. PLoS One 2020;15(12):e0243693
2. Mishra T, Wang M, Metwally AA, Bogu GK, Brooks AW, Bahmani A, et al. Pre-symptomatic detection of COVID-19 from smartwatch data. Nat Biomed Eng 2020 Dec;4(12):1208-1220.
3. Hirten RP, Danieletto M, Tomalin L, Choi KH, Zweig M, Golden E, et al. Use of physiological data from a wearable device to identify SARS-CoV-2 infection and symptoms and predict COVID-19 diagnosis: observational study. J Med Internet Res 2021 Feb 22;23(2):e26107
4. Hasty F, García G, Dávila CH, Wittels SH, Hendricks S, Chong S. Heart rate variability as a possible predictive marker for acute inflammatory response in COVID-19 patients. Mil Med 2020 Nov 18:e34
5. Hajduczuk AG, DiJoseph KM, Bent B, Thorp AK, Mullholand JB, MacKay SA, Barik S, Coleman JJ, Paules CI, Tinsley A
Physiologic Response to the Pfizer-BioNTech COVID-19 Vaccine Measured Using Wearable Devices: Prospective Observational Study
JMIR Form Res 2021;5(8):e28568

Reviewer #2 (Remarks to the Author):

The study of Gepner et al. reports the result of an exciting application of wearables for the continuous assessment of physiological measures to investigate the short-term effects of the BNT162b2 COVID-19 vaccine.

The study is well conducted and explained and tries to overcome the actual limitations posed by the subjective and self-reported questionnaires. I have several comments and suggestions that I hope will help to improve the paper and strengthen the interesting findings reported. The study is well conducted and explained and tries to overcome the actual limitations posed by the subjective and self-reported questionnaires. I have several comments and suggestions that I hope will help to improve the paper and strengthen the reported findings .

The core of the study is based upon the aggregate analysis of vitals and parameters estimated by a commercial wearable device. This chest-patch medical device allows estimating 13 physiological indicators. The device received the FDA approval for the heart rate (ECG trace), respiratory rate, SYS and DIA blood pressure values. In order to use all the other parameters especially in the analysis I expect to see valid references with clinical trials aiming at validating this device. Otherwise, I suggest to comment the results but explicitly report the absence of validity studies in the limitations section. I'm a little dubious about the validity of the data during daily activities. Is the device also validated

for artifacts caused by body movements? If I understand correctly this patch is intended for hospital monitoring, where such movements are really very small. In case it is not validated, I invite the authors to clarify this point and to consider only the data taken in the night phase.

In line 236 Authors reported "By contrast, we observed no significant differences in blood oxygen saturation (Fig. 1B)." How the device measure the blood oxygen saturation at the level of the chest? How reliable is that measurement. The same for the temperature. I suggest to report the body temperature as skin temperature, if the skin temperature is the one measured with the device.

In Table I, I suggest to report all the measurement units.

I didn't understand what the two apps are for. If I understand correctly, one app is for passively collecting data from the wearable device; the other PerMed app is for filling out the daily questionnaire. If so, the sentence in line 149 doesn't make much sense "[...] the PerMed mobile application passively collects smartphone sensory data, as well as allows participants to fill the daily questionnaires."

Moreover, I am not very clear on the choice of questions presented within the PerMed App. Why not ask more questions, why a scale of 1-5?

I am very unclear about the 4 day monitoring option. Couldn't a change in parameters happen even beyond the third day after the vaccine (last day monitored with the device)?

I do not understand then why the control subjects (first dose) are monitored only 2 days (Fig S8) and did not instead follow an experimental protocol as that of the subjects with second dose (subject of the study).

Reviewer #1:

Gepner et al. present a report of a wearable monitoring system that records physiologic data and the changes in those physiologic variables in response to the COVID-19 vaccine. The study design is sound and the manuscript does not have any major pitfalls. It is, however, disorganized as it reports results/discussion prior to the study population and study design. These should be re-organized.

Response: We thank the reviewer for the comment. We reordered the manuscript such that the Methods section now comes before the Results and Discussion sections.

While it is clear that the authors allocated a great deal of resources to this report, it is unclear to me how this would inform additional studies, from a physiologic perspective, outside of showing the feasibility of this study design/use of the wearable monitor. The physiologic metrics, which are the main strength of the paper and heavily analyzed in the results, need to be discussed in greater detail in regards to their importance and how this can be leveraged to either determine vaccine response, antibody titers or protective immunity.

Response:

We agree with the reviewer and are thankful for this point. We, therefore, added the following paragraph to the Discussion section:

"Clinical trial guidelines for assessing the safety of vaccines, including the FDA criteria, are primarily based on subjective, self-reported questionnaires. Using a real-time remote patient monitoring platform that can provide multiple objective physiological parameters has significant implications for clinical studies of all phases in learning about adverse events, safety, and choosing the ideal treatment protocol. Moreover, it has been recently suggested that reactions caused by the COVID-19 vaccine are a byproduct of a short burst of IFN-I generation concomitant with the induction of an effective immune response²⁰. Thus, future studies should evaluate the association between physiological reactions and vaccine effectiveness."

The wellness and other subjective outcomes are fair to report, but they do not add much to the overall impact of the manuscript, aside from providing external validation of the much larger studies already published. The authors went to great lengths to collect this data with the mobile app, but even a profound impact on subjective outcomes may not even be clinically significant in the setting of an efficacious vaccine to end a global pandemic.

Response: We agree with the reviewer and therefore decided to remove (entirely) the wellness part from the manuscript.

The next issue with this manuscript is the definition of the term "baseline." The device was worn for a total of 4 days (1 day prior to vaccination and 3 days after). The authors claim a "return to baseline," which is an inaccurate statement as they did not determine a true baseline for these physiologic metrics. This essentially nullifies any claims made about a return to baseline, as many of these factors fluctuate on a day-to-day basis and a "change from baseline" should be calculated based on at minimum a 30-day dataset (if not more).

Response: We agree with the reviewer regarding the use of the term "baseline". To address this difficulty, we replaced the term 'baseline' with the more accurate wording 'day prior vaccination' throughout the manuscript.

A strength of the manuscript is its prospective nature and size (n=160). The latter is worth highlighting as it is likely one of the larger studies with vaccines and wearables, specifically the COVID vaccine. In addition, the descriptive analysis of the systemic and local reactogenicity was well done, and overall the figures are easy to follow.

Response: We highly appreciate the reviewer's comments that helped us improve the manuscript. The sample size is presented in the Abstract. We now added the following sentence into the discussion:

"... compared to other studies monitoring vaccines with wearable sensors, and particularly COVID-19 vaccines, our sample size is relatively large."

Specific comments:

1. Recommend including the brand name of the vaccine that was used in this study for the lay person in the manuscript title. And please comment on whether the authors feel these results would be generalizable across all mRNA COVID-19 vaccines.

Response: We agree with the reviewer that the paper title should be more specific. As most academic papers do not explicitly write the commercialized brand name (i.e., Pfizer BioNTech), we changed the title so that it now includes the medical name of the vaccine:

"Utilizing wearable sensors for objective, continuous and highly-sensitive monitoring of reactions to the BNT162b2 mRNA COVID-19 vaccine".

With regard to generalization to other COVID-19 vaccines we added the following sentence to the Discussion section:

"Given the similarities in the extents and the types of local and systemic reactions observed between different COVID-19 vaccines^{8,11,12}, we believe our findings are likely to be qualitatively similar in other vaccine types."

2. Title should specify that this report focuses on COVID-19 vaccination. While this may be applicable to all vaccines, it is not within the scope of this manuscript to make this claim (can be touched upon in the discussion).

Response: We made sure to include in the updated title the term "COVID-19 vaccine".

3. Per above, please organize the manuscript to have the study population/design/methods etc before the results and discussion (should follow the outline of the abstract).

Response: We reordered the manuscript as suggested - i.e. the Methods section now appears before the Results and Discussion sections.

4. Was sleep duration a measured variable with the monitoring device? If so, please report.

Response: Unfortunately, sleep duration is not monitored by the chest-patch sensor.

5. The Day 0 and daily changes (days 1-3) in variables should be reported in table format to show the inter and intraindividual variability seen in the cohort.

Response: Thank you. We now added a table to the supplement materials with all daily measures (Table S1). Please note, due to IRB guidelines, we are not allowed to share the raw data at the individual level. However, we now provide for each measure the mean and std. observed in each day and night.

6. Heart rate variability (HRV) has been studied as marker of automatic function and perturbation, and correlations between HRV and vaccine responses have been reported in the literature. More recently, HRV has been studied in several reports related to COVID-19 vaccine (citations below). These should certainly be cited in this manuscript and commented upon in the discussion (specifically, why the authors think their continuous HRV measurements may not have shown as robust of a change than other reports in the literature).

1. Miller DJ, Capodilupo JV, Lastella M, Sargent C, Roach GD, Lee VH, et al. Analyzing changes in respiratory rate to predict the risk of COVID-19 infection. PLoS One 2020;15(12):e0243693
2. Mishra T, Wang M, Metwally AA, Bogu GK, Brooks AW, Bahmani A, et al. Pre-symptomatic detection of COVID-19 from smartwatch data. Nat Biomed Eng 2020 Dec;4(12):1208-1220.
3. Hirten RP, Danieleto M, Tomalin L, Choi KH, Zweig M, Golden E, et al. Use of physiological data from a wearable device to identify SARS-CoV-2 infection and symptoms and predict COVID-19 diagnosis: observational study. J Med Internet Res 2021 Feb 22;23(2):e26107
4. Hasty F, García G, Dávila CH, Wittels SH, Hendricks S, Chong S. Heart rate variability as a possible predictive marker for acute inflammatory response in COVID-19 patients. Mil Med 2020 Nov 18:e34
5. Hajduczuk AG, DiJoseph KM, Bent B, Thorp AK, Mullholand JB, MacKay SA, Barik S, Coleman JJ, Paules CI, Tinsley A

Physiologic Response to the Pfizer-BioNTech COVID-19 Vaccine Measured Using Wearable Devices: Prospective Observational Study

JMIR Form Res 2021;5(8):e28568

Response: Following the reviewer's comment, we added the following paragraph to the Discussion section which extends the literature review with the suggested references and highlights the importance of HRV:

"Several recent studies highlighted the pivotal role of Heart Rate Variability for the detection of COVID-19 infection ²¹⁻²⁵. Specifically, one study showed that a combination of measures from smartwatches, including HRV, can be utilized to detect COVID-19 during the pre-symptomatic phase ²². Although significantly changed, we did not observe a robust signal from the HRV measure following vaccination. As HRV is reported to be an index of the influence of both the parasympathetic nervous system and the sympathetic nervous systems ²⁶, it could be that vaccination affects these systems differently than infection."

Reviewer #2 (Remarks to the Author):

The study of Gepner et al. reports the result of an exciting application of wearables for the continuous assessment of physiological measures to investigate the short-term effects of the BNT162b2 COVID-19 vaccine.

The study is well conducted and explained and tries to overcome the actual limitations posed by the subjective and self-reported questionnaires. I have several comments and suggestions that I hope will help to improve the paper and strengthen the interesting findings reported. The study is well conducted and explained and tries to overcome the actual limitations posed by the subjective and self-reported questionnaires. I have several comments and suggestions that I hope will help to improve the paper and strengthen the reported findings.

Response: We thank the reviewer for finding our study exciting, well-conducted and explained, and we highly appreciate the thoughtful comments.

The core of the study is based upon the aggregate analysis of vitals and parameters estimated by a commercial wearable device. This chest-patch medical device allows estimating 13 physiological indicators. The device received the FDA approval for the heart rate (ECG trace), respiratory rate, SYS and DIA blood pressure values. In order to use all the other parameters especially in the analysis I expect to see valid references with clinical trials aiming at validating this device. Otherwise, I suggest to comment the results but explicitly report the absence of validity studies in the limitations section. I'm a little dubious about the validity of the data during daily activities. Is the device also validated for artifacts caused by body movements? If I understand correctly this patch is intended for hospital monitoring, where such movements are really very small. In case it is not validated, I invite the authors to clarify this point and to consider only the data taken in the night phase.

Response:

The chest patch sensor received CE Mark approval for all 13 measures (CE 2797) (Biobeat Technologies Ltd). Regarding body movement, the device has an inherent component that prevents showing values if the movement has crossed a pre-set threshold.

To increase the validity of the chest patch sensors in the paper, we revised the relevant paragraph in the Methods section to include additional citations:

“The chest patch sensor received FDA clearance for measures of heart rate, blood oxygen saturation, and systolic and diastolic blood pressure (clearance number K190792) and CE Mark approval for all 13 measures (CE 2797) (Biobeat Technologies Ltd), (see also a technological validation study for blood pressure¹⁵). The sensor tracks vital signs derived from changes in the pulse contour, following calibration using an approved non-invasive, cuff-based device, and is based on Pulse Wave Transit Time (PWTT) technology, combined with Pulse Wave Analysis (PWA) (see, for example,^{16,17}). To increase its validity, the device has an inherent component that prevents showing values if the movement has crossed a pre-set threshold. To the best of our knowledge, this is the only cleared wearable wireless medical-grade device to provide all these measurements.”

In line 236 Authors reported "By contrast, we observed no significant differences in blood oxygen saturation (Fig. 1B)." How the device measure the blood oxygen saturation at the level of the chest? How reliable is that measurement. The same for the temperature. I suggest to report the body temperature as skin temperature, if the skin temperature is the one measured with the device.

Response: The device received CE Mark approval. SpO2 is measured from the chest using the same PPG-based sensor that measures the other cardio-pulmonary parameters. We updated the text to refer to the temperature as 'skin temperature'.

In Table I, I suggest to report all the measurement units.

Response: Thank you. We added the measurement units to the tables and in the text when describing the 13 indicators in the Methods section.

I didn't understand what the two apps are for. If I understand correctly, one app is for passively collecting data from the wearable device; the other PerMed app is for filling out the daily questionnaire. If so, the sentence in line 149 doesn't make much sense "[...] the PerMed mobile application passively collects smartphone sensory data, as well as allows participants to fill the daily questionnaires."

Response: We now understand that the sentence might be misleading. We did not use the sensory data from the PerMed app – i.e., we used only the daily questionnaires. To prevent confusion, we deleted the confusing part. The revised sentence now reads as follows:

"... the PerMed mobile application allows participants to fill the daily questionnaires"

Moreover, I am not very clear on the choice of questions presented within the PerMed App. Why not ask more questions, why a scale of 1-5?

Response: Thank you. Following the suggestion of Reviewer 1, we decided to remove (entirely) the questions regarding the wellbeing indicators.

I am very unclear about the 4 day monitoring option. Couldn't a change in parameters happen even beyond the third day after the vaccine (last day monitored with the device)?

Response: Due to technical difficulties (the battery of the chest-patch sensor lasts for 4-5 days and budget constraints) individuals carried the chest-patch sensors for 4 days. We would like to emphasize, however, that individuals filled the daily questionnaire for a longer period of 14 days and that the vast majority of the participants did not show symptoms after 2 days following vaccination (figure S4). For transparency, we added the following sentence to the Methods section:

"As the chest patch sensors' battery typically lasts for 4-5 days, participants were asked to remove the chest patch sensors three days after vaccination."

I do not understand then why the control subjects (first dose) are monitored only 2 days (Fig S8) and did not instead follow an experimental protocol as that of the subjects with second dose (subject of the study).

Response: This is, indeed, a mistake. Thank you for your input. We now updated Figure S8 to include the measures for the third day as well.

REVIEWERS' COMMENTS:

Reviewer #1 (Remarks to the Author):

The authors made substantial changes, based on the reviewer comments, that strengthened the manuscript. I have no further requests on the updated version.

Reviewer #2 (Remarks to the Author):

The paper has been extensively revised accordingly to my comments. I have no further questions for the authors. The paper can be now accepted.